

# A quantitative comparison of towed-camera and diver-camera transects for monitoring coral reefs

Anna K. Cresswell[1,2], Nicole M. Ryan[1], Andrew J. Heyward[1], Adam N. H. Smith[3], Jamie Colquhoun[1], Mark Case[1], Matthew J. Birt[1], Mark Chinkin[1], Mathew Wyatt[1], Ben Radford[1,2], Paul Costello[4] and James P. Gilmour[1,2]

[1] Australian Institute of Marine Science, Perth, WA, Australia
[2] Oceans Institute, University of Western Australia, Perth, WA, Australia
[3] School of Natural and Mathematical Sciences, Massey University, Auckland, New Zealand
[4] Australian Institute of Marine Science, Townsville, QLD, Australia

Corresponding author
Anna K. Cresswell,
a.cresswell@aims.gov.au

## ABSTRACT

Novel tools and methods for monitoring marine environments can improve efficiency but must not compromise long-term data records. Quantitative comparisons between new and existing methods are therefore required to assess their compatibility for monitoring. Monitoring of shallow water coral reefs is typically conducted using diver-based collection of benthic images along transects. Diverless systems for obtaining underwater images (e.g. towed-cameras, remotely operated vehicles, autonomous underwater vehicles) are increasingly used for mapping coral reefs. Of these imaging platforms, towed-cameras offer a practical, low cost and efficient method for surveys but their utility for repeated measures in monitoring studies has not been tested. We quantitatively compare a towed-camera approach to repeated surveys of shallow water coral reef benthic assemblages on fixed transects, relative to benchmark data from diver photo-transects. Differences in the percent cover detected by the two methods was partly explained by differences in the morphology of benthic groups. The reef habitat and physical descriptors of the site—slope, depth and structural complexity—also influenced the comparability of data, with differences between the tow-camera and the diver data increasing with structural complexity and slope. Differences between the methods decreased when a greater number of images were collected per tow-camera transect. We attribute lower image quality (variable perspective, exposure and focal distance) and lower spatial accuracy and precision of the towed-camera transects as the key reasons for differences in the data from the two methods and suggest changes to the sampling design to improve the application of tow-cameras to monitoring.

PeerJ ___________________________________________

## INTRODUCTION

Marine mapping and monitoring programs need accurate and precise assessments of benthic communities. An increased focus on monitoring coral reefs followed the first global coral bleaching event in 1998, given the unprecedented scale and severity of its impacts (*Houk & Van Woesik, 2013*). Coral reefs have since been the subject of some of the most extensive monitoring in the marine environment (*Mellin et al., 2020*). Many coral reef monitoring programs involve recording the percent cover of corals and other benthic groups from photographs obtained by scuba divers, usually collected along a transect line, i.e. 'photo-transects' (*Littler et al., 1997*). The photos are then processed by placing random or fixed points on the images, identifying the benthic organism under the points, and converting this to an estimate of percent cover (*Jonker, Johns & Osborne, 2008*). Comparisons of marine benthic communities among locations and through time depend on consistency in survey design, scale and methods (*Brown et al., 2004*; *Edmunds, 2013*; *Galzin et al., 2016*; *Jonker, Johns & Osborne, 2008*; *Obura et al., 2017*).

Marine monitoring programs usually require expensive and time-consuming field work, particularly at reefs that are remote or difficult to access, making it important to allocate survey effort effectively (*McDonald-Madden et al., 2010*). Advances in technology offer opportunities for collection of image-based data via novel methods that may be safer, more efficient or economical, enable surveys over larger areas, or allow for increased replication and greater statistical power (*Houk & Van Woesik, 2006*; *McDonald-Madden et al., 2010*). Furthermore, as acute disturbances to coral reefs increase in spatial scale and frequency (*Hughes et al., 2018*), rapid, responsive monitoring is important. However, changes in methods increase variability in data and can compromise long-term data series (*Bicknell et al., 2016*; *Durden et al., 2016*). Thus, there is ongoing discussion regarding the best methods for coral reef monitoring (*Brown et al., 2004*; *Houk & Van Woesik, 2013*; *Lam et al., 2006*; *Leujak & Ormond, 2007*; *Mellin et al., 2020*).

Visual imaging, using photos or video, is an essential component of most marine monitoring methods as it is non-destructive, provides a permanent record and is relatively efficient in the field (*Carleton & Done, 1995*; *Durden et al., 2016*; *Flannery & Przeslawski, 2015*). Image-collection platforms such as remotely operated vehicles (*Raoult et al., 2020*), automated underwater video (*Foster et al., 2014*; *Ludvigsen & Sørensen, 2016*) and towed-cameras (*Carroll et al., 2020*; *Davis et al., 2019*; *Sheehan et al., 2016*) using digital stills or video continue to become more sophisticated and economical. Among these remote systems, towed-cameras are a practical and efficient method for mapping reefs (*Barker et al., 1999*; *Carleton & Done, 1995*; *Carroll et al., 2020*; *Cruz-Marrero et al., 2019*; *Davis et al., 2019*; *Lembke et al., 2017*) but diver-based collection of images along transects remain the standard for monitoring reefs, with demonstrated ability to capture changes through time at fine spatial scales.

As demonstrated by the evolution from in situ visual census to image-capture in marine monitoring in the mid-1990s (*Carleton & Done, 1995*), new survey methods will inevitably be introduced into coral reef monitoring programs as technology evolves. Choice of method and survey design have important implications for the effectiveness, cost and

interpretation of monitoring efforts (*Boavida et al., 2016*; *Houk & Van Woesik, 2013*; *Jokiel et al., 2015*; *Mellin et al., 2020*). The method employed should reflect the broader aims of the monitoring program while also considering the scale and taxonomic resolution of the information required, the repeatability of a method and environmental conditions. For example, the ability of towed-camera systems to capture informative seabed imagery can be influenced by sea conditions, water clarity, and the depth and type of terrain being surveyed (*Carroll et al., 2020*; *Durden et al., 2016*).

Understanding how the accuracy and precision of data from new methods compares to the historical benchmarks of diver-based transects is a critical step in assessing their suitability (*Flannery & Przeslawski, 2015*; *Lam et al., 2006*). There are various comparisons of diver-based methods with other sampling techniques for fishes (*Boavida et al., 2016*), but relatively few have focused on benthic invertebrates, algae or benthic communities (*Boavida et al., 2016*; *Brown et al., 2004*; *Foster et al., 2014*; *Jokiel et al., 2015*; *Lam et al., 2006*; *Leonard & Clark, 1993*), and there are no quantitative comparisons of towed-camera systems and diver photo-transect methods for monitoring coral reef habitats, despite their increasing application. Here, we compare surveys of shallow (<12 m) coral reefs using a small tow-camera system with diver-operated camera surveys on fixed transects in four distinct habitats. This study aims to: (1) determine the accuracy and precision of the towed-camera system in following a fixed-position transect; (2) compare the benthic communities and coral assemblages detected by each method; (3) identify any factors—specifically habitat, depth, substratum complexity or reef slope (gradient)—that explain differences between the two methods; (4) describe the utility and evaluate the best approach to using towed-camera systems for monitoring shallow coral reefs.

## MATERIALS AND METHODS

### Study site

The Rowley Shoals is a group of three isolated oceanic shelf reefs near the edge of the continental shelf ~260 km from mainland north west Australia (Fig. 1). The reefs—named Mermaid, Clerke and Imperieuse—are separated from each other by >25 km of oceanic water (*Wilson, 2013*).

Benthic surveys were conducted in March 2018 and October 2019 at 10 permanently marked monitoring sites (Fig. 1). Sites were stratified by four habitats: 'slope' on the reef slope at ~6 m depth (at lowest astronomical tide, LAT); 'crest' on the reef crest at ~3 m LAT, both on the outer eastern boundary of Mermaid, Clerke and Imperieuse reefs; 'lagoon floor' at the base of coral bommies in the lagoon at Clerke and Mermaid reefs (9–12 m LAT); and 'bommie' on the top of coral bommies in the lagoon at Clerke and Mermaid reefs (~1 m LAT) (Fig. 1). Surveys were prioritized during slack tides, when winds were below 20 knots and swell was less than 1 m.

Permits to conduct these surveys were received from the Australian Government Department of the Environment (Mermaid Reef Commonwealth Reserve), the Western Australian Department of Primary Industries and Regional Development (DPIRD #2999) and the Western Australian Department of Biodiversity, Conservation and Attractions.

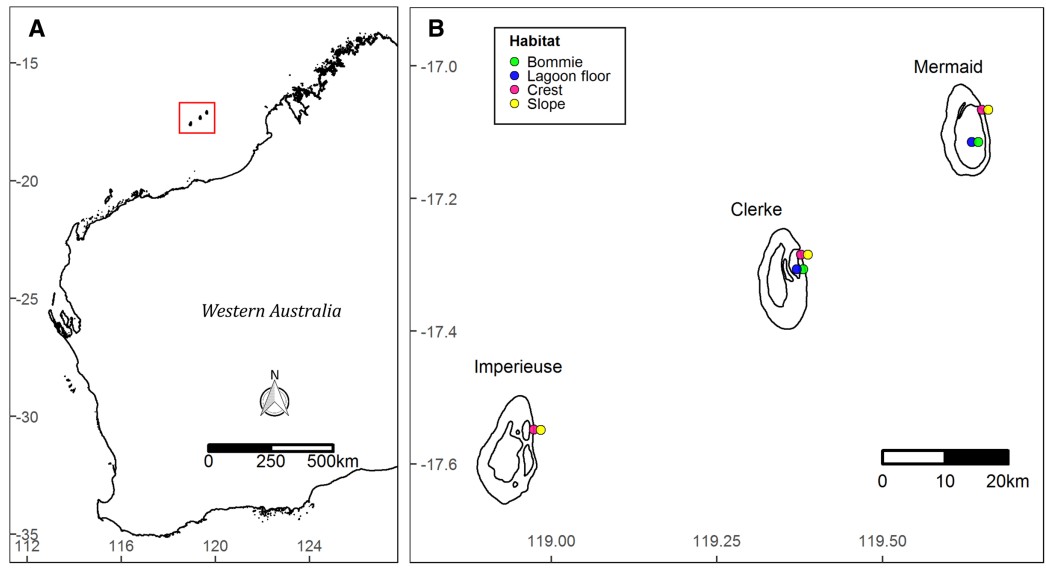

**Figure 1** **Site map.** Location of the Rowley Shoals in northwest Western Australian (A). Study sites and habitats (B).

## Survey methods

At each site there were fixed continuous transects, each separated by approximately 10 m. The total length of transects was 250 m on the reef slope (composed of five 50 m-long transects), 150 m on the reef crest (three 50 m-long) and 120–150 m at the lagoon floor and coral bommie sites (three 50 m-long in 2018 and six 20 m-long in 2019) (Fig. 2). Reef crest and slope transects ran along depth contours parallel to the reef crest, while the lagoon floor and bommie transects wrapped around the base or top of the bommies respectively (Figs. S1.3 and S1.4).

Sites were surveyed on scuba following the standard operating procedures (SOP) of the AIMS long-term monitoring program (*Jonker, Johns & Osborne, 2008*). The dive team required 2–3 personnel (photographer and dive buddy, surface attendant if needed). The divers used a digital Canon PowerShot G1 X Mark II (12.8 MP, 1.5 inch sensor) in an underwater housing, photographing the substrate every metre along the transects, with the camera held perpendicular ~50 cm from the substratum. The resulting images each capture an area of approximately 23 × 30 cm. It took ~45–60 min to collect images along five 50 m end-to-end transects, depending on the substratum complexity of the reef and the water conditions, such as surge or current. After the images were collected, a geo-referenced track of the transects was obtained by a swimmer following the transect line at the surface with a GPS.

The 'tow-camera' was designed to be hand deployable from a tender or small vessel for the specific purpose of surveying shallow reef areas that have traditionally been surveyed by divers. The system is therefore lightweight, battery powered, and much smaller than more traditional tow-video systems (e.g. *Carroll et al., 2020*), measuring 25 × 27 × 75 cm. The tow-camera system consists of an aluminum frame, to which an oblique-angled high-definition Sony analogue video camera is attached, providing a live video feed via a

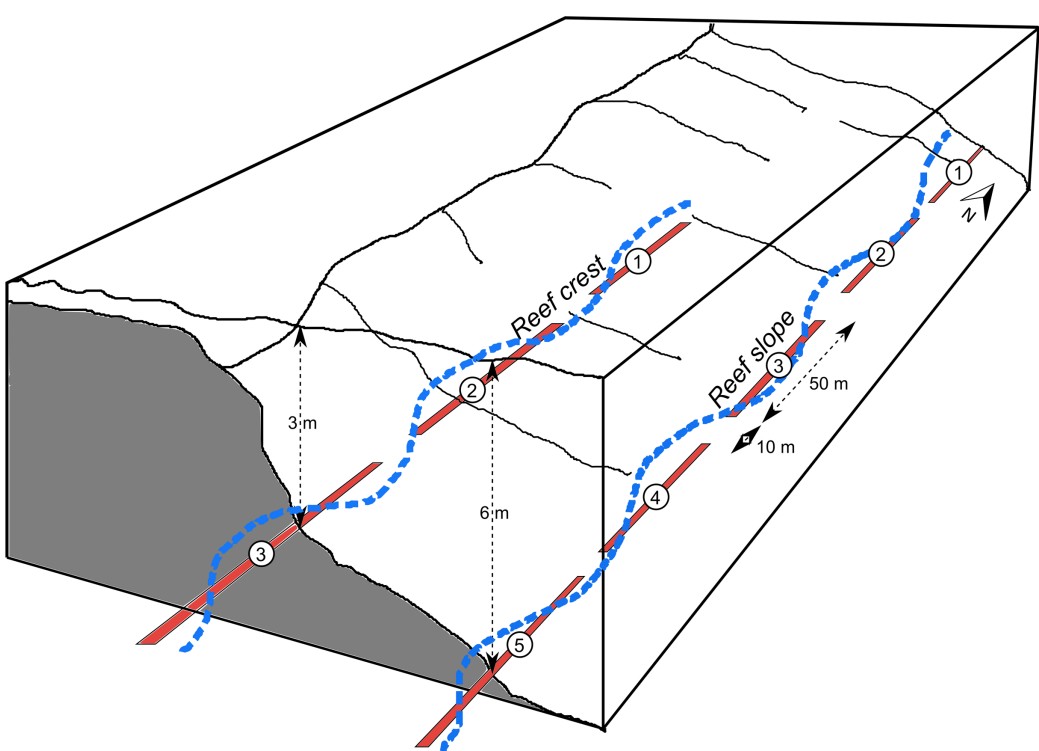

**Figure 2 Schematic of transects.** Schematic showing examples of fixed, 50 m-long diver transects (red), separated by 10 m, for a reef crest and reef slope site. A continuous GPS track of these transects was followed using the tow-camera with images continuously collected every 3 s (e.g. blue track), a process which was replicated 2–4 times. Diagram modified from *Sweatman et al. (2005)*.

30 m loadbearing cable to a monitor (Atmos) on the boat. A downward-facing Panasonic Lumix DMX-LX100 Mk I (13 MP, Four Thirds sensor) digital camera, was set to photograph at 3 s intervals with an INON Z240 strobe (Fig. S1.1). The tow-camera was deployed from a 6 m rigid hulled inflatable boat and minimum of two people conducted the surveys: one adjusted the height of the system to approximately 0.5 m from the substratum while viewing the live video feed (*Bowden & Jones, 2016*; *Carroll et al., 2020*) and one drove the boat along the GPS track of the diver transects at a speed of 1–2 knots. Slower speeds were preferable, but the boat needed to move fast enough to stay on track. The optimum time to complete a 50 m transect, assuming the boat maintained exact course along the transect, would be 2.5 min, corresponding to a speed of 0.33 ms$^{-1}$ (0.65 knots) and with a photo interval of 3 s, one photo per 1 m. Each transect was repeated 2–4 times, to increase image replication, so total survey time for each site was approximately 30–60 min. Images were geo-referenced based on the image timestamp matched to the GPS on the boat, which gave an approximation for the position of the tow-camera, which followed behind the boat at approximately 5–10 m. We estimate the distance of the tow-camera from the substrate varied between 25 and 175 cm, meaning the area captured in an individual image likely varied between approximately 13 × 18 cm and 81 × 108 cm.

## Image selection

The tow-camera continuously photographed (every 3 s) along the full length of all transects at a site, so individual transects were not distinguished at the time of the tow (Fig. 2). Two to four replicates were collected by turning the boat and repeating the same track over the transects. The geo-location of each tow-image was then used to classify each image into one of the transects; polygons were created from the GPS diver track in QGIS to represent each transect with a gap between (see Fig. S1.2). Any images that were in the gap between transects and those collected when the boat was turning around were removed. Images from replicated tows were pooled into their respective transects based on image position.

The shortest distance between each tow-camera image and the diver GPS track was measured in metres using 'distance to nearest hub' in QGIS Desktop 3.10.0. The mean and range of distances from the diver track were calculated for each tow and any tow-camera images more than 5 m from the diver track were removed from further analyses. On the diver transects, one image was taken every metre, i.e. 50 images per 50 m transect, or 20 images per 20 m transect (Table 2). For the tow transects, the total number of images varied depending on the number of images within the 5 m buffer surrounding the diver track, the speed of the boat and the number of replicate tows at a site.

## Image analysis and benthic categories

Two observers experienced in coral reef taxonomy analysed the images using point sampling, where the substratum under each of five fixed points overlaid on each image was assigned to the finest taxonomic resolution that could be confidently identified (see *Jonker, Johns & Osborne, 2008*); for corals this was usually to genera.

Benthic data were classified and analysed at two taxonomic scales: (i) the 'benthic community', which grouped all hard corals together and considered the proportions of abiotic substratum, macroalgae, turf/coralline algae and soft corals as separate categories, as well as the proportions of unidentified substratum and points that did not have sufficient image quality to be identified; and (ii) the hard 'coral community'. A combination of genera, family and growth form described the coral groups to best represent the distinctive groups present at the Rowley Shoals (see Table 1).

## Covariates

We were interested in whether the slope, depth, or substratum complexity accounted for differences between the data from the two methods (Table 2). We used laser airborne depth sounder data (LADS, https://www.navy.gov.au/ran-aviation-history/laser-airborne-depth-sounder-lads), to calculate the mean and standard deviation of depth along the transects, and to estimate substratum complexity. The LADS point cloud was interpolated to form a raster surface using a unidirectional and optimised universal kriging model in the Geostatistical Analyst module in ArcGIS™ version 10.6. The final raster resolution was 2.66 m² pixels which was a stable estimation of the variogram model sill point and finer

**Table 1 Categories used to describe the benthic communities and the coral communities at the Rowley Shoals.**

| Category | Description and occurrence at Rowley Shoals |
|---|---|
| **Benthic community** | |
| Crustose coralline algae | Crustose coralline algae and fine turf algae, suitable for colonisation by coral recruits |
| Macroalgae | Large fleshy algae, which are rare across the reef system, but if present can exclude and outcompete coral recruits |
| Soft coral | *Lobophytum* and *Sinularia*. Found at all three reefs but most common at Mermaid Reef |
| Hard coral | Scleractinian corals and *Millepora* (fire coral) |
| Abiotic substratum | Sand, rubble, dead coral, shells |
| Other organisms | Ascidians, hydroids (not *Millepora*), non-mobile gastropods, zoanthids |
| Sponge | Relatively rare at the depths studied |
| Unidentified substratum | No benthic ID was possible from the point on the photo due to image quality or obstruction by diver, tape or mobile organisms; Image very blurry or not directed at the substratum (e.g. only shows water) |
| **Coral community** | |
| Corymbose and digitate *Acropora* | Corymbose and digitate growth forms common across the Rowley Shoals |
| Tabular *Acropora* | Tabulate growth forms common across the Rowley Shoals |
| Branching *Acropora* | Branching growth forms common across the Rowley Shoals |
| *Astreopora* | Rare, but some present on the Mermaid lagoon floor |
| *Diploastrea* | Characteristic of Mermaid Reef |
| Foliose corals | *Echinopora, Merulina, Pachyseries, Pectinia, Turbinaria* |
| Fungiidae | Rare, but some present at Mermaid lagoon floor. *Fungia* |
| *Isopora* | *Isopora brueggemanni* and *I. palifera* |
| Merulinidae | *Goniastrea, Coelastrea, Dipsastrea* (formerly *Favia*), *Favites, Cyphastrea* and other Merulinidae species, mostly with a massive growth form. Corals from the genus *Leptastrea* (*L. insertae, L. sedis*) are also included in this group. |
| Encrusting corals | Encrusting forms of *Montipora, Acanthastrea, Acropora, Galaxea, Merulina, Pectinia, Psammocora* and *Turbinaria* |
| *Pavona* | Including encrusting and submassive growth forms |
| Pocilloporidae | *Pocillopora, Seriatopora, Stylophora* |
| *Porites* | Mostly massive forms, with branching morphology rare |
| *Millepora* | Hydrozoa within the Family Milleporidae |
| Soft corals | *Lobophytum* and *Sinularia*. Found at all three reefs but most common at Mermaid Reef |
| Uncommon (grouped for univariate analyses, included as separate groups in multivariate analyses) | Other hard coral genera not included above that were present but had less than a maximum of 2% cover across all eight sites such that we would not expect them to be reliably detected by either survey method (*Coeloseris, Galaxea, Gardineroseris, Goniopora, Hydnophora, Lobophyllia, Merulina, Montastrea, Oulophyllia, Pachyseris, Pectinia, Plerogyra, Plesiastrea, Psammocora, Symphyllia, Turbinaria*) |
| Unidentified coral | A coral, but classification into one of the above groups was not possible from the image |

scale interpolation would have introduced stochastic error. From the same LADS raster, we used the ArcGIS™ surface analysis tool in Spatial Analyst to calculate the mean slope of each transect, as the maximum rate of change between each cell and its neighbours in degrees.

We also considered whether there was a relationship between the number of tow-camera images per metre and the difference between the two methods, to test whether more tow-camera images increased the accuracy of the tow-camera relative to the diver.

**Table 2 Covariates considered in statistical models. Transformations are given in square brackets.**

| Variables (transformation) | Degrees of freedom | Values/range | Fixed or random |
|---|---|---|---|
| Method | 1 | *Diver* or *Tow* | Fixed |
| Habitat | 3 | *Slope, Crest, Lagoon floor, Bommie* | Fixed |
| Reef | 2 | *Mermaid, Clerke, Imperieuse* | Fixed |
| Survey year | 2 | *2018* or *2019* | Fixed |
| Depth (m) | 1 | 0.35–11 | Fixed |
| Substratum complexity | 1 | 0.06–2.83 | Fixed |
| Slope [$\log_e$] | 1 | 2.3–45.6 | Fixed |
| Number of images (m$^{-1}$) [$\log_e$] | 1 | *Tow*: Mean 2.5, range 0.3–3.4 *Diver*: Mean 1, range 1–1 | Fixed |

## Multivariate statistical analyses

Benthic and coral community assemblage data were analysed by transect using multivariate and univariate methods. The percent cover of each benthic group was calculated from the number of points assigned to each group in the images divided by the total number of points per transect. All multivariate analyses were conducted in PRIMERv7 (PRIMER-e) using Bray–Curtis dissimilarities calculated from square-root transformed data (*Clarke & Gorley, 2015*). Community assemblage data were visualised with non-metric multidimensional scaling (nMDS) (*Clarke & Gorley, 2015*) to examine potential separation between groups associated with the four main factors of interest: method, habitat, reef and year. Vector overlays of Pearson correlations >0.6 were used to identify the most influential benthic and coral groups driving separation in the data. Additional vector overlays with Pearson correlations >0.15 were used to visually assess the extent to which the physical covariates—depth, depth variation or slope—were associated with separation in the data cloud. These physical variables were transformed in cases where their distribution was particularly right skewed: slope and substratum complexity were $\log_e$ transformed.

We used permutational multivariate analysis of variance (PERMANOVA; *Anderson, 2001*) to test whether method, habitat, reef, survey year, or two-way interactions between these factors affected the composition of benthic and coral communities. Higher order interactions were excluded. Models were reduced by excluding two-way interactions that had a $p$-value > 0.25 before re-running the analysis. Residuals were permutated (9,999 permutations) under a reduced model, and tests were based on Type III (partial) sums of squares. We tested the hypothesis that the ability of the tow-camera to detect similar benthic and coral community assemblages to the diver method may vary between habitats and therefore specifically examined pairwise tests on the interaction of habitat and method.

## Univariate statistical analyses

Differences in the percent cover detected between methods of each of the broad benthic groups and hard coral taxa were investigated using linear models in R
(*R Development Core Team, 2018*). The analysis was conducted at the transect level and designed to investigate whether mean depth, slope, substratum complexity, habitat and/or number of tow-camera images per transect correlated with the difference in the percent cover between the two methods, $\Delta\text{cover}_{i,j}$, where

$$\Delta\text{cover}_{i,j} = \text{cover diver}_{i,j} - \text{cover tow}_{i,j}$$

and $i$ is the benthic group and $j$ is the transect in question.

$\Delta$cover was modelled as the response variable and habitat, reef and year were included as fixed effects. Site was not included directly in the model, because there was only one site per combination of these fixed effects, and therefore the combination of these factors determined the sites indirectly, thus accounting for spatial autocorrelation of transects. The response variable $\Delta\text{cover}_{i,j}$ was bound by a minimum of $-100\%$ and a maximum of $100\%$, and a Gaussian error model was used. Homogeneity of variance was checked visually by plotting the residuals against the fitted values. Residuals were examined and were satisfactory for conforming to the normal distribution assumption in most cases. For the coral taxa groups that only appeared on a handful of transects—*Diploastrea*, Fungiidae, *Pavona* and unidentified coral—the residuals were not normally distributed, and we did not have confidence in these models, and therefore did not include these statistics. While we may have considered a zero-inflated Gaussian model, or presence-absence models, we did not because there were so few occurrences with which to provide any robust conclusions about these groups. Collinearity among the continuous predictor variables was assessed: slope and substratum complexity were highly correlated ($r > 0.75$) and therefore only slope was included in the models. Predictor variables were transformed to reduce the skewness where appropriate: the slope and the number of images m$^{-1}$ were log$_e$-transformed (Table 1). A full model containing all variables was fit first and sequentially simplified based on Akaike Information Criterion.

We focused on the intercept of the models when predictor variables were at their mean to test whether there was a significant difference between the two methods under average conditions. Therefore, we mean-centred the continuous variables—number of images, slope and depth—and used sum-to-zero contrast matrices for the categorical variables habitat, year and reef. The model intercepts were directly interpreted as $\Delta\text{cover}_i$ under average conditions for continuous variables and marginalised over the factors.

In interpreting the results, a mean positive difference in cover ($\Delta\text{cover}_i > 0$) meant the diver images detected a higher cover of $i$ than the tow-camera images. Conversely, a mean negative difference in cover ($\Delta\text{cover}_i < 0$) meant the diver images detected a lower cover of $i$ than the tow-camera images.

## RESULTS

The mean distance of the boat towing the camera at the time of image acquisition was 2–3 m either side of the diver transect GPS track (see Figs. S1.3 and S1.4; Table S1.1). Images were excluded from analyses if the boat deviated more than 5 m from the diver GPS track (30% of 2,226 images from the 2018 survey and 22% of 4,194 images from the

**Table 3 PERMANOVA partitioning and analysis of the benthic (eight categories) and coral community (16 categories) assemblages at the Rowley Shoals, based on square-root transformed percent cover and Bray–Curtis dissimilarities.**

| Source | df | SS | MS | Pseudo-F | p | Component | Variance | Sq.root |
|---|---|---|---|---|---|---|---|---|
| Benthic community | | | | | | | | |
| ME | 1 | 2,370.0 | 2,370.0 | 27.79 | **0.0001** | Fixed | 35.7 | 6.0 |
| HAB | 3 | 17,597.0 | 5,865.6 | 68.79 | **0.0001** | Fixed | 153.3 | 12.4 |
| RE | 2 | 3,737.1 | 1,868.6 | 21.91 | **0.0001** | Fixed | 38.4 | 6.2 |
| YR | 1 | 761.4 | 761.4 | 8.93 | **0.0002** | Fixed | 9.4 | 3.1 |
| ME × HAB | 3 | 3,295.0 | 1,098.3 | 12.88 | **0.0001** | Fixed | 53.4 | 7.3 |
| ME × RE | 2 | 650.2 | 325.1 | 3.81 | **0.0031** | Fixed | 9.7 | 3.1 |
| ME × YR | 1 | 2,823.9 | 2,823.9 | 33.12 | **0.0001** | Fixed | 68.9 | 8.3 |
| HAB × RE | 4 | 7,933.1 | 1,983.3 | 23.26 | **0.0001** | Fixed | 118.2 | 10.9 |
| RE × YR | 2 | 573.8 | 286.9 | 3.36 | **0.0059** | Fixed | 7.9 | 2.8 |
| Res | 146 | 1,2450 | 85.3 | – | – | Random | 85.3 | 9.2 |
| Total | 165 | 57,200 | – | – | – | – | – | – |
| Coral community | | | | | | | | |
| ME | 1 | 568.5 | 568.5 | 1.18 | 0.3489 | Fixed | 1.1 | 1.0 |
| HAB | 3 | 71,538.0 | 23,846.0 | 49.47 | **0.0001** | Fixed | 643.8 | 25.4 |
| RE | 2 | 17,302.0 | 8,651.0 | 17.95 | **0.0001** | Fixed | 172.9 | 13.1 |
| YR | 1 | 1,311.6 | 1,311.6 | 2.72 | **0.0141** | Fixed | 11.2 | 3.3 |
| ME × HAB | 3 | 3,445.1 | 1,148.4 | 2.38 | **0.0016** | Fixed | 32.8 | 5.7 |
| HAB × RE | 4 | 17,878.0 | 4,469.4 | 9.27 | **0.0001** | Fixed | 246.7 | 15.7 |
| HAB × YR | 3 | 4,091.1 | 1,363.7 | 2.83 | **0.0001** | Fixed | 45.7 | 6.8 |
| Res | 148 | 71,347 | 482.1 | – | – | Random | 482.1 | 22.0 |
| Total | 165 | 205,000 | – | – | – | – | – | – |

**Note:**
'Source', sources of variation in the model; 'df', degrees of freedom; 'SS', sums of squares; 'MS', mean squares; 'Pseudo-F' is the pseudo F-ratio and 'p', permutation P-value. 'Variance' estimates sizes of components of variation based on multivariate analogues to the classical ANOVA unbiased estimators. 'Sq.root' gives the square root of these values, so is in Bray–Curtis units. In the Source column, ME, method; HAB, habitat; YR, survey year; RE, reef. Significant effects are indicated in bold.

2019 survey). The density of remaining images ranged from between 0.3 and 3.4 images m$^{-1}$, with a mean of 1.3 (0.8 SD) m$^{-1}$, which was more than that of the diver surveys (1 m$^{-1}$).

## Benthic community structure

Variation in benthic community structure was explained by the method of sampling, habitat, reef, survey year and interactions between these factors (Table 3). The main effect of method was significant, indicating that, over and above the interactions, there was an overall difference in the benthic community structure detected between the tow-camera and diver methods (Table 3). The greatest components of variation (a measure of the magnitude of the effect) were for the interaction between reef and habitat, and habitat as a main effect (Table 3). A significant method-by-habitat interaction indicates that the differences between the methods varied among the four habitats. Significant differences in the benthic community composition exisited between the sampling methods within all

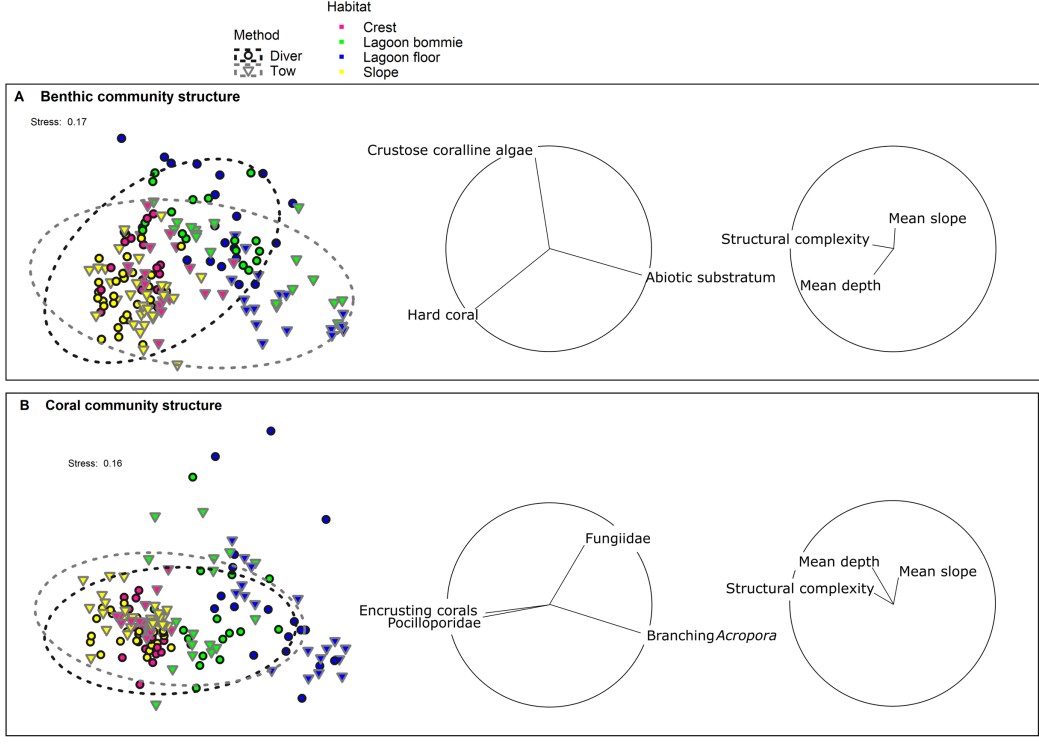

**Figure 3 nMDS of (A) the benthic community structure and (B) the coral community structure.**
Black outlined circles represent the diver transects whereas the downwards facing grey triangles represent tow-camera transects. The dashed ellipses indicate 95% spatial confidence intervals for the centroids of the two methods with overlap therefore indicating similarity in the assemblages identified. Colour identifies the four habitats surveyed. Vectors with Pearson correlations >0.6 and >0.15 for biological and environmental variables respectively are shown to the right of the nMDS plots to indicate the main benthic/coral groups and depth, slope and structural complexity parameters causing separation in the data cloud.

four habitats, but to different extents (Table S3.1). Among the habitats, the benthic communities separated mainly due to differences in the cover of hard corals; in particular, the reef crest and slope had more hard corals than the lagoon floor and bommies (Fig. 3A). Across all transects, the tow-camera method recorded a lower percent cover of crustose coralline algae and hard coral, a higher cover of abiotic substratum, and a higher proportion of images that could not be assigned to a group due to poor image quality ('unidentified substratum').

Correlations between the site covariates (slope, substratum complexity, depth) and the multivariate ordination axes were weak (Fig. 3A). Univariate analyses indicated that steeper slope (correlated with substratum complexity) was associated with the tow-camera underestimating the cover of hard corals, while obtaining a higher cover that could not be identified (Table S3.4; Figs. 4 and 5A). Increases in depth correlated with the tow-camera recording lower cover of crustose coralline algae, and more abiotic substratum relative to the diver method. The two methods became more comparable in their estimates of hard corals as depth increased (Fig. 5B).

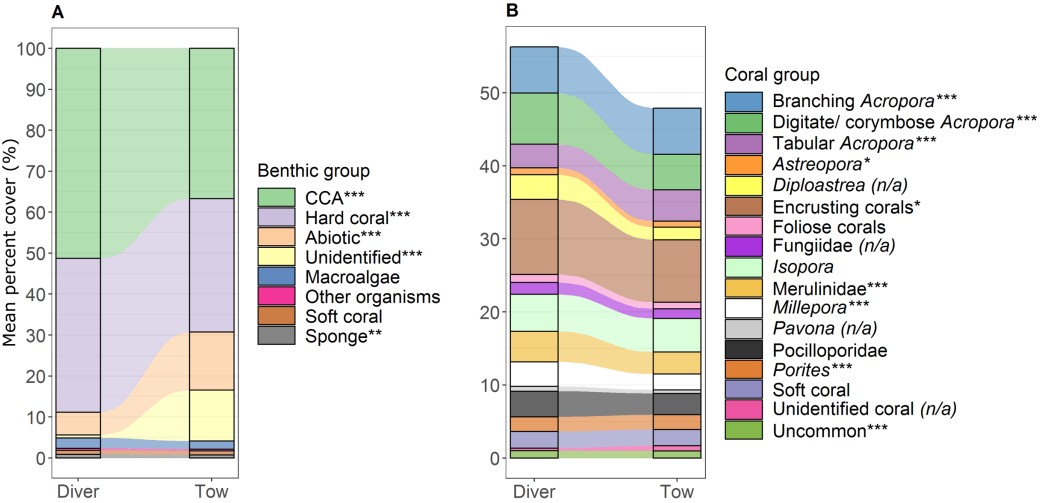

**Figure 4 Comparison of percent cover between methods.** Stacked bar plots with blocks representing the mean percent cover of each (A) benthic and (B) coral group for the diver and the tow-camera survey methods, with stream fields between the bars illustrating the difference between the methods. Asterisks in legend indicate a significant difference between the two methods as identified from linear models when all predictor variables were at their mean (***$p < 0.001$, **$p < 0.01$, *$p < 0.05$; (n/a) is specified when the data did not meet the assumptions for robust linear modelling). See Table S.3.2 for associated mean and standard error values.           

## Coral community structure

The coral community structures identified by the two methods were more similar than the broader benthic community structures (Fig. 3; Table 3). As expected, habitat was the most important correlate of variation in coral community structure, but there was also a significant interaction between habitat and method (Pseudo-F = 2.38, df = 3, $p = 0.0016$; Table 3), indicating differences between the methods depended on the habitat. Pair-wise tests showed no differences in the coral assemblages identified by the two methods at the lagoon floor ($t = 1.10$, df = 32, $p = 0.32$) and bommie sites ($t = 1.18$, df = 30, $p = 0.25$), but significant differences at the reef crest ($t = 1.95$, df = 31, $p < 0.001$) and reef slope sites ($t = 2.33$, df = 55, $p < 0.001$; Table S3.1).

The most notable differences in the mean cover of coral taxa between methods was a lower cover of encrusting corals, *Isopora*, Merulinidae, *Millepora* and digitate and corymbose *Acropora* detected by the tow-camera, and a lower cover of tabular *Acropora* in the diver images (Fig. 4; Table S3.4). Differences in the cover of coral taxa between methods varied among habitats (Fig. 6), reefs and survey years (Table S3.4). For example, the tow-camera surveys detected lower cover of encrusting hard corals relative to the diver surveys at bommie sites but more cover of this group at the lagoon floor sites (Fig. 6; Table S3.4). The diver surveys tended to detect more Merulinidae, *Millepora*, Pocilloporidae and *Porites* than the tow-camera (Fig. 4; Table S3.4) as slope increased and more Merulinidae and digitate and corymbose *Acropora* as depth increased (Fig. 5).

For several benthic and coral groups, an increase in the number of images per metre increased the comparability of the tow-camera data to the diver data (hard coral: $t = 0.72$,
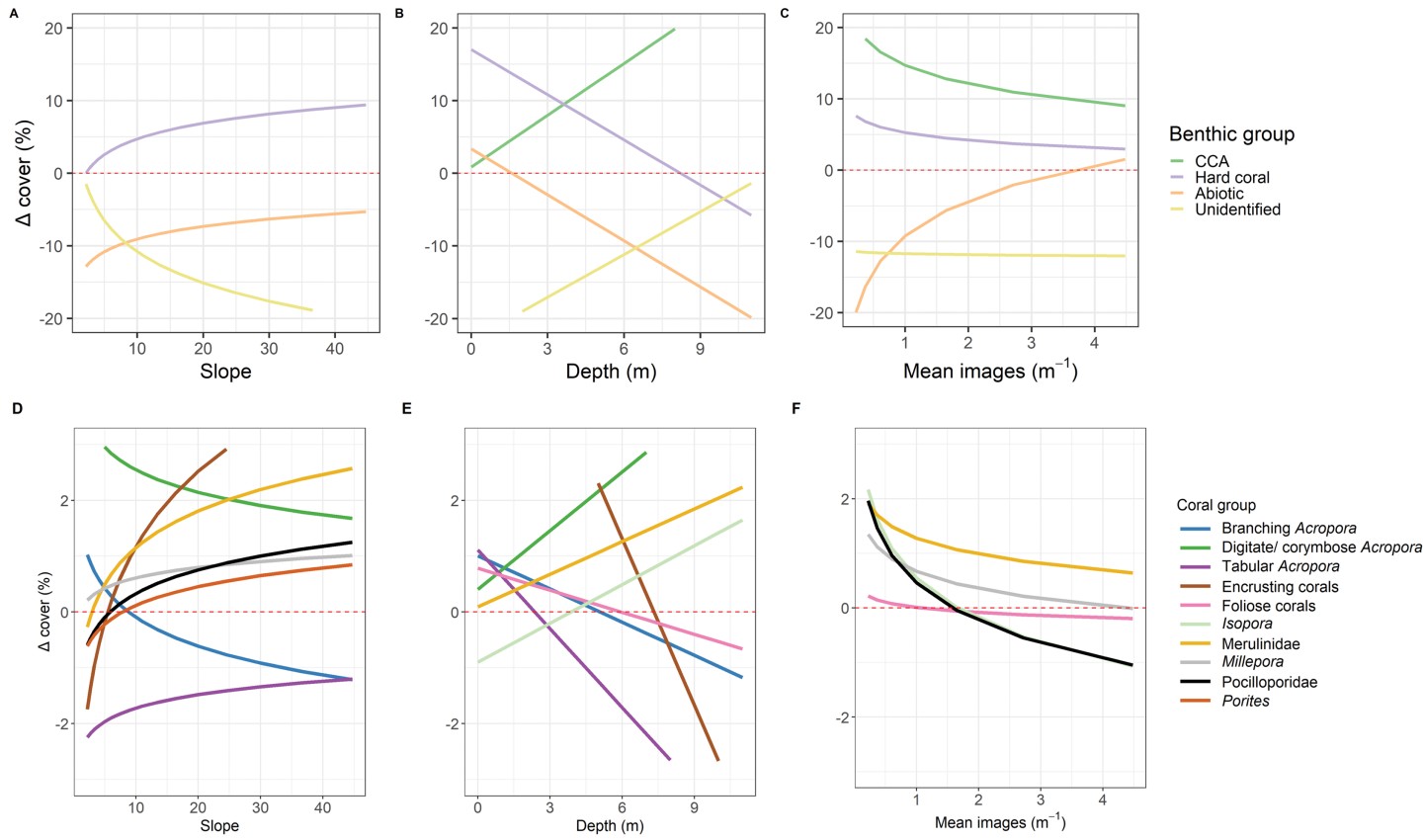

**Figure 5 Predictions from linear models.** Predictions from the top linear models for the common benthic (A–C) and coral (D–F) groups. Marginal effects of each continuous variable at the mean of the factor variables are shown to indicate how these variables relate to the difference between the two survey methods where Δcover$_i$ is calculated as diver—tow-camera. Only significant relationships are shown here (see Table S3.4).

$p < 0.001$; crustose coralline algae: $t = 0.83$, $p < 0.001$; abiotic substratum: $t = 0.86$, $p < 0.001$; Merulinidae: $t = 0.19$, $p = 0.028$; *Millepora*: $t = 0.21$, $p = 0.030$; Figs. 5C and 4F; Table S3.4).

## DISCUSSION

We assessed the suitability of a small towed-camera system for monitoring coral reefs relative to the benchmark of diver photo-transects. The ability of the tow-camera to produce comparable data to diver photo surveys depended on the benthic groups of interest, the physical conditions of survey sites and the number of tow images collected. We attribute image quality (perspective, exposure and focal distance) and limits to the spatial accuracy and precision of the towed-camera system as the two key drivers of differences in the data obtained from the two methods. Our quantitative investigation allows informed suggestions on the best approach to using towed-camera systems for monitoring.

A combination of distance of the tow-camera from the substratum and images not being taken perpendicular to the seafloor likely explain the generally negative influence of slope and substratum complexity on the performance of the tow-camera system. Greater substratum complexity required the operator to adjust the depth of the tow-camera

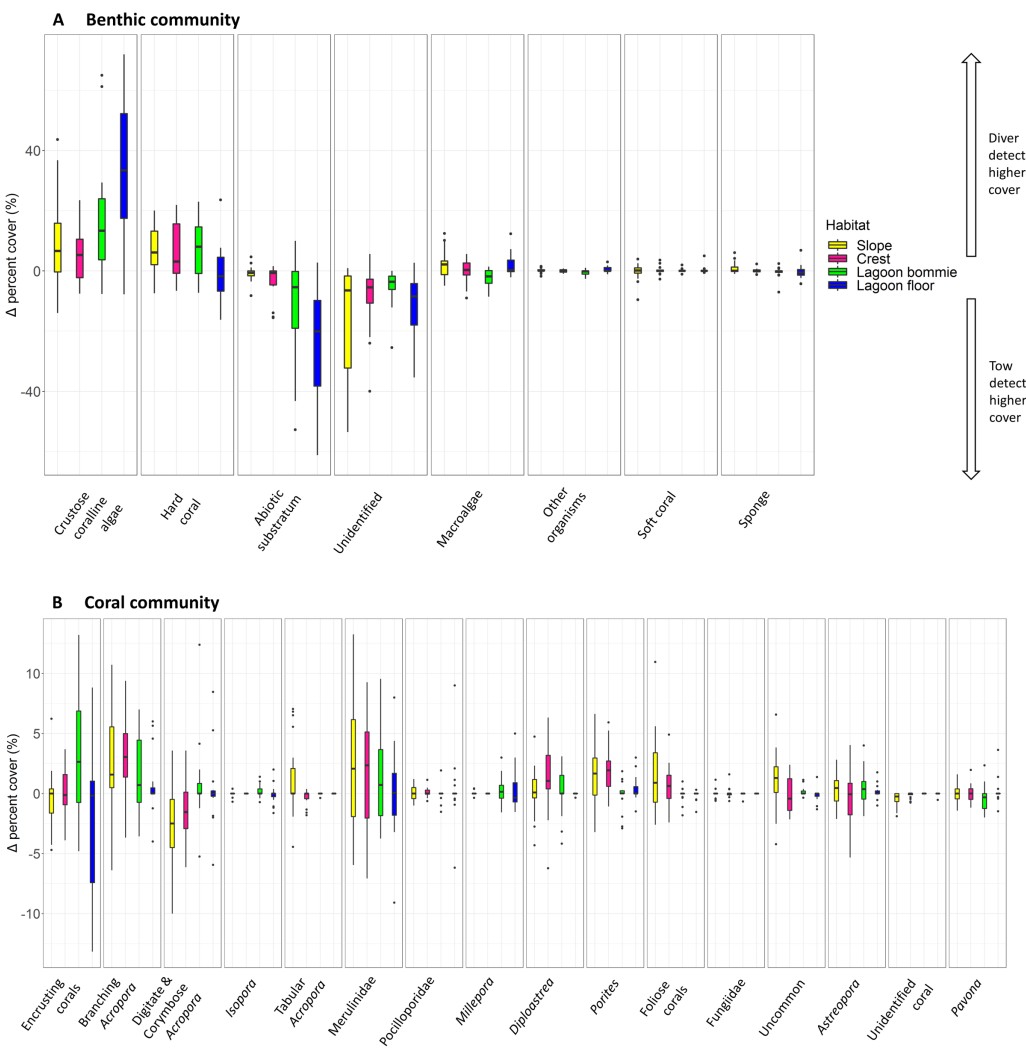

**Figure 6 Boxplots of habitat differences.** Boxplots of the difference in percent cover between the two methods (diver—tow-camera), $\Delta \text{cover}_i$ , for each group of the (A) benthic and (B) coral community categories for each habitat ($n_{slope}$ = 30 paired, all other habitats $n$ = 18 paired transects).

more often, leading to more variability in the image quality. Previous studies have also established the challenges of using towed cameras in environments with high rugosity or rough terrain (*Durden et al., 2016*). The distance between the camera and the substratum will clearly influence image quality and the identification of benthic and coral groups, with diver images collected at a consistent distance of approximately 50 cm from the substratum, but more variability in the tow-camera images (we estimate 25–150 cm) (*Bicknell et al., 2016*; *Davis et al., 2019*; *Leonard & Clark, 1993*). Our estimates of depth, substratum complexity and slope in the present study were coarse and fine-scale measurements of these variables along each transect would have allowed greater insight into the link between tow-camera image quality and the physical attributes of sites. Generally however, the substratum was identifiable to the taxonomic resolution examined in the present study, though identifications from images could take longer when images

were poor. Camera systems configured with fast shutter speeds and/or larger depth of field could improve the image quality from the tow-camera. Automatic image stabilization systems and other evolving technologies will likely improve image capture ability, facilitating use of larger image sensors that can offer greater dynamic range and higher resolutions, embedding more information in the captured images.

Estimates of benthic community structure differed more between the two methods than for coral assemblages, which can be partly attributed to the morphologies of the benthic and hard coral groups. The tow-camera provided lower estimates of crustose coralline algae and encrusting hard corals, but higher estimates of abiotic substratum. These groups have less distinctive shapes and colours, and less three-dimensional structure than other benthic and coral groups, making them more difficult to distinguish in tow-camera images of lower quality. There were differences in the percent cover of *Acropora* growth forms detected by the two methods, possibly reflecting the tow images capturing a greater area per image (greater distance from substratum) and thus detecting more corals with overtopping morphologies, while underestimating encrusting and small or cryptic colonies. *Bowden & Jones (2016)* have highlighted that camera orientation is a critical parameter for quantitative interpretation of imagery. In comparing downward- vs. forward-facing diver-operated video, *Bennett et al. (2016)* found more vertically erect corals and algae, but less turf algae in the forward-facing video. In our study, the diver collected images were near-perpendicular to the seafloor, capturing a planar view, whereas there was more variation in the tow-camera image perspective, perhaps meaning 3D forms were more often resolved.

Maneuvering a vessel to remain on a specified GPS track at low speeds (<2 knots) while keeping a towed camera tracking along the seabed was more challenging in higher wind and swell conditions, while physical descriptors of the sites (slope, substratum complexity and depth) also influenced image quality. Images >5 m either side of the GPS track for each transect were removed from analyses, creating a transect width of ~10 m for the tow-camera system, i.e., a much wider transect than the 1–2 m of the diver transects. Therefore, we would expect differences in the data from the two methods simply due their surveying different areas of seafloor. Substantial fine-scale variation and zonation in benthic assemblages can occur over distances of less than 5 m on coral reefs (*Storlazzi et al., 2016*; *Underwood & Chapman, 1996*). As spatial heterogeneity and gradients of community zonation increase, the ability to precisely resurvey the same area becomes more important and can have a large effect on statistical power (*Ryan & Heyward, 2003*). Indeed, in the present study, a 5 m deviation perpendicular from a lagoon bommie transect could take the tow-camera onto the lagoon floor, or a similar deviation from a reef slope transect could take the camera onto the reef flat, or further down the slope into much deeper (~20 m) water. Another source of variation was the position of the tow-camera relative to the boat, as the position of each image was determined by the location of the boat rather than the tow-camera system, creating additional imprecision in spatial location of the images. A better understanding of where images were collected can be obtained by calculating the position of the camera relative to the boat based on the length and angle of the cable extended (see *Carroll et al., 2020*), but this does not improve the

ability to track along a transect. Coral assemblages detected by the two methods were similar in the lagoon habitats (lagoon floor and bommies) but were significantly different on the reef crest and slope. This may be because the reef slope and crest habitats were more difficult to survey due to the environmental conditions (more exposed) and/or because the coral assemblages are more diverse and can vary over finer spatial scales in these habitats than in the lagoon habitats.

The divers took approximately 45–60 min to complete five end-to-end 50 m transects, requiring a minimum team of two people, while the tow team required the same minimum number of people and a comparable amount of time (30–60 min) as transects were repeated 2–4 times to increase replication. The cost of equipment, maintenance and preparation for surveys, were similar for both methods: a setup for the diver-camera may be in the order of $2 K, with additional costs for diver equipment up to $8 K, whereas the cost for the tow-camera system is $7–10 K. The tow-camera is not limited by diver safety restraints, both in terms of the safety of the environment being surveyed (e.g. dangerous marine animals, current, depth) or constraints on dive time. The image analysis tended to be slower for the tow-camera images than the diver images, with suboptimal exposure and focus in some images making it more difficult to distinguish benthic groups quickly, however artificial intelligence may speed the image processing time for both methods in the future (*González-Rivero et al., 2020*).

## CONCLUSIONS

Tow-camera systems offer an alternative approach to monitoring shallow water coral reefs, but with important trade-offs and situation- and location-dependent considerations. The same level of spatial accuracy and precision as a diver on a fixed transect cannot be achieved with the tow-camera, making it difficult or impossible to monitor at a level where the same communities are likely to be revisited in repeated sampling. In the present study, sampling with the tow-camera was constrained by applying the same sampling design as was used in the diver surveys, to make comparisons between the methods as rigorous as possible. However, we suggest the tow-camera would be better applied to monitoring with a dedicated sampling design reflecting the strengths and weaknesses of the system. Given the spatial variability when attempting to follow a linear transect with the tow-camera, and the result that an increase in the number of images improved compatibility with the diver data, we suggest more heavily sampling (more images) an area that defines a region of interest. Variations of the tow-camera system used in the present study have been used to sample large areas (1 km$^2$) to produce benthic habitat maps and models for shallow and deep coral reefs (*Heyward & Radford, 2019*). For monitoring, variability and zonation in habitats are key considerations for a revised sampling design as this approach would be of limited success if the area being sampled spanned different habitats (e.g. reef slope to reef crest; bommies to lagoon floor) or was highly heterogeneous. The size of the survey area is therefore important; for example *Jokiel et al. (2015)* showed that longer transects (25 and 50 m) were useful for relatively homogeneous substratum but shorter transects (10 m) were more appropriate in heterogeneous habitats. Therefore, a

georeferenced polygon within habitats of interest in the order of 0.1–1.0 hectares in area, large enough to maneuver a small boat within, might be most successful.

We would suggest any established diver-based long-term monitoring programs should maintain continuity in methods where possible, given the tow-camera could not achieve the same results, with implications for sampling design and statistical power. Nonetheless, the tow-camera is an alternative for surveying shallow reef environments when it is not possible for divers to enter the water. The comparison presented here will allow informed interpretation of future data collected by tow-cameras and a clearer understanding of the strengths, weaknesses and biases relative to diver-camera data.

## ACKNOWLEDGEMENTS

We would like to thank Diego Barneche and Rebecca Fisher for discussions on statistical analyses; the crew and field teams on the RV Solander on both the field trips when this data was collected, in particular Kim Brooks; the workshop team in Townsville for maintaining the tow-camera system and Neill Roberts and Melanie Olsen for writing a Standard Operating Procedure for this tow-camera, and providing the schematic shown in an appendix. An earlier version of the manuscript benefited from comments provided by Karen Miller, Scott Bainbridge and Vicki Nelson.

### Funding

This study was conducted as part of AIMS' North West Shoals to Shore Research Program and was supported by Santos as part of the company's commitment to better understand Western Australia's marine environment. The funders had no role in study design, data collection and analysis, decision to publish, or preparation of the manuscript.

### Grant Disclosures

The following grant information was disclosed by the authors:
Santos.

### Competing Interests

The authors declare that they have no competing interests.

### Author Contributions

- Anna K. Cresswell analyzed the data, prepared figures and/or tables, authored or reviewed drafts of the paper, and approved the final draft.
- Nicole M. Ryan conceived and designed the experiments, performed the experiments, analyzed the data, authored or reviewed drafts of the paper, and approved the final draft.
- Andrew J. Heyward conceived and designed the experiments, analyzed the data, authored or reviewed drafts of the paper, key contributor to the creation, design and ongoing improvement on the tow-camera system, and approved the final draft.
- Adam N. H. Smith analyzed the data, authored or reviewed drafts of the paper, and approved the final draft.

- Jamie Colquhoun conceived and designed the experiments, performed the experiments, authored or reviewed drafts of the paper, and approved the final draft.
- Mark Case conceived and designed the experiments, performed the experiments, authored or reviewed drafts of the paper, and approved the final draft.
- Matthew J. Birt conceived and designed the experiments, performed the experiments, authored or reviewed drafts of the paper, and approved the final draft.
- Mark Chinkin conceived and designed the experiments, performed the experiments, authored or reviewed drafts of the paper, and approved the final draft.
- Mathew Wyatt analyzed the data, authored or reviewed drafts of the paper, and approved the final draft.
- Ben Radford analyzed the data, authored or reviewed drafts of the paper, and approved the final draft.
- Paul Costello analyzed the data, authored or reviewed drafts of the paper, and approved the final draft.
- James P. Gilmour conceived and designed the experiments, performed the experiments, analyzed the data, authored or reviewed drafts of the paper, and approved the final draft.

## Field Study Permissions

The following information was supplied relating to field study approvals (i.e., approving body and any reference numbers):

Field experiments were approved by the Department of Biodiversity Conservation and Attractions and Department of Primary Industries and Regional Development (DPIRD #2999).

## Data Availability

Raw data is available in the Supplemental Files.

## Supplemental Information

Supplemental information for this article can be found online at http://dx.doi.org/10.7717/peerj.11090#supplemental-information.

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
