# Peer review of "A quantitative comparison of towed-camera and diver-camera transects for monitoring coral reefs"

_PeerJ, doi:10.7717/peerj.11090_

## Round 0.1 · original submission · Minor Revisions

As you will see below, both our referees are quite enthusiastic about your work and offer some suggestions for possible improvement. In particular, Referee #1 raises some questions about your statistical approach that should be answered in your rebuttal.

Please ensure that all review comments are addressed in a rebuttal letter and any edits or clarifications mentioned in that letter are also inserted into the revised manuscript where appropriate. It is a common mistake to address reviewer questions in the rebuttal letter but not in the revised manuscript. If a reviewer raised a question, then your readers will probably have the same question so you should ensure that the manuscript can stand alone without the rebuttal letter. Directions on how to prepare a rebuttal letter can be found at: https://peerj.com/benefits/academic-rebuttal-letters/

I look forward to seeing your revised manuscript.

Reviewer 1 ·

Basic reporting

The manuscript is very well presented and is easy to read and follow. I do provide some suggestions and recommendations list below.

Much of the introduction mentions costs involved with survey methods but there is no mention about the difference in costs of running diver survey verses a towed video survey. I would suggest, maybe in the discussion, having a few lines that state a towed video system cost $xx and cost $xx to survey xx sites versus $xx dollars for a diver survey. If there is a significant cost difference this could help a perspective reader decided what method is appropriate etc. I have seen some method comparisons papers provide a table that includes cost of unit, number of people required for survey, cost a days survey etc. Could you even provide a decision tree / flow chat for the discussion/conclusion?

Experimental design

- My main question around the study design and statistical analysis is around the dreaded pseudo-replication. I can see two potential issues of pseuda-replicaiton and wonder what the authors think or if it should be addressed. Neither of these are deal breakers, especially as the objective is to compare two methods. However, I still think they should be consider?
o The first, including method as a fixed factor in the PERMANOVA? I understand they have included the interaction between method and habitat etc. However, PERMANOVA assumes independence of all samples. Is a sample from towed video is taken from the exact same location as a diver survey surely, they are not independent? Therefore, the risk of type 1/2 error is a concern? I would have thought doing 2 separate PERMANOVAs, one for each method, and testing to see if the results are within x% of each other and therefore able to make the same inference regardless of the method?
o The second, having individual transects going end to end ie. 5 transects in a 250m site? I know this is a commonly use method but it also doesn’t necessarily make it correct. A review of the Victorian Marine Park monitoring program found this level of pseudo-replication was an issue and had to change the methodology. Has this been tested? Does transect or site need to be considered as a random factor to account for spatial correlation?
- Line 314 you mention that differences between methods varied with habitat? Is this true variability or could this be due to a low sample size? Is it worth doing a power analysis to see if both methods have the ability to detect change?

Validity of the findings

No comment

Additional comments

Minor comments
- Line 32-33 - this sentence is confusing I would suggest rewording.
- Line 34 - percent cover of what?
- Line 43 – I would like to see a concluding sentence in the abstract. Something along the lines of a recommendation i.e we recommend towed video can replace diver survey if a monitoring program is focussed on….. At the moment I am left wondering what the authors are recommending until the very end of the discussion. I always suggest never leave the reader hanging.
- Line 50 – this doesn’t just apply to conservation planning, it also applies to spatial management, impact assessment, fisheries management, oil and gas exploration etc.
- Line 62 – how are the data comprimise?
- Line 65 – Ongoing discussion about what?
- Lines 62 and 65 don’t really link the paragraphs. I would almost remove the opening sentence on Line 65.
- Line 70 – Technical advances also allow for an increase in replication, greater sample size = (sometimes) greater abilitiy to detect change = greater statistical power.
- Line 75 – you can also add it provides a permanent record.
- Line 77 – This could be a useful reference: https://www.frontiersin.org/articles/10.3389/fmars.2019.00134/full
- Line 74-79 – Have you seen the NESP Field Manuals? It would be worth making reference to this collection of field manuals/ SOPs as they were design to ensure that there are standardised method to allow for detection of change, reproducibility, comparability etc etc. Most importantly there is a Towed Video, ROV, AUV chapter.
- Line 82- I don’t think this is true? Towed video/cameras/ROV have been used in monitoring programs for decades. I have been using these technologies for 20 years now! What I think has changed is that they are now more affordable, more options, more accessible to the average researcher / low budget projects etc.
- Line 125-126 – Slight rewording to improve reading, suggest “… and due to their isolation and conservation protection they have not experience many….
- Line 162 – include the megapixels and sensor size given that you mention this for the towed video system.
- Line 275 – Why did you fit a model with a Poisson, ZIP or negative binomial distribution that would account for what I assume is zero inflation? Are these species of interest? Often it is the less common rare species that are the most important for detecting changing in a monitoring program? Are they worth including?
- Line 304 – I would also include the standard error after the mean.
- Line 370 – Can I suggest a closing sentence along the lines of towed video is comparable to diver surveys…. Could be used to replace diver surveys etc. Once again as the reader I am left hanging till the conclusions to know what you would recommend.
- Discussion in general, so is camera quality that is most important? Do we need to test multiple camera brands, types, lighting etc to find a more suitable towed video camera?
- Line 423 – This more future suggestion comment that you may already be aware of, in the past I have use a camera – boat offset to geo-locate the photo with the GPS track. I bit of simple Pythagoras theorem, length of cable out + depth and you can calculate how far the camera is behind the GPS.
- Conclusions, I’m still hang to know if towed video could replace diver surveys. The results suggest it depends on many factors. If I was a funding body or a manager I want to know that after 10 years of monitoring can I swap to towed video and get usable comparable data? This is why I think a decision try would be useful?
- Figure 3 – Can I suggest changing a method symbol to circle or square or even open. Having opposite triangles with so many sample is hard for the reader to disentangle the patterns, especially when a color scale is involved.

Reviewer 2 ·

Basic reporting

Interesting contribution. Well written manuscript with an elegant data analysis statistical approach.

Experimental design

Methods: presented with logical, well designed statistical design. Thorough Results and the Discussion.

Validity of the findings

Although the Structure for Motion/Photogrammetry survey techniques are revolutionizing coral reef monitoring, it will not be long before technology replaces swinging divers by towed-camera systems able to follow the bottom contour. This manuscript provides a baseline for future forthcoming protocol updates and technology advancement.

Additional comments

Cresswell et al. A quantitative comparison of towed-camera and diver-camera transects for monitoring coral reefs

General comments:
Interesting contribution. Well written manuscript with an elegant data analysis statistical approach. As authors indicate, with limited budgets, increasing operational costs, and limited staff, remote sensing techniques have rapidly gained traction among coral reef monitoring programs worldwide. Although the work presented in this is not particularly innovative, its scientific merit and dissemination power are worth publication.
Methods: presented with logical, well designed statistical design. Thorough Results and the Discussion. Although the Structure for Motion/Photogrammetry survey techniques are revolutionizing coral reef monitoring, it will not be long before technology replaces swinging divers by towed-camera systems able to follow the bottom contour. This manuscript provides a baseline for future forthcoming protocol updates and technology advancement.

Just a few comments section by section:

Methods:

Lines 124-132: Seems to me that this is unnecessary detail, much of which is irrelevant to the objectives of the MS. I suggest authors limit the site description to the location of the reefs.

Line 135: “Sites were stratified by habitat: ‘forereef slope’ …. ‘reef crest’. Seems to me these habitat names are self-explanatory and do not necessitate the added description.

Lines 140-149: Streamline…. “Based on the prevailing conditions, general surveys were prioritized during slack tides. Winds below 20kn and swell less than 1 m. All survey activities were permitted accordingly”.

-What was the area of each image photographed by divers and towed-camera; was the area compatible?

Line 200-201: How did select fixed points? Arbitrarily/stratified? Why only 5 points per image? Did you experiment with different number of points on the TDC images? An increase in the number of points could potentially increase the accuracy and eliminate a portion of the unidentified due to image blur?

Line 269: Zuur et al 2010 absent in References

Line 236: In my experience, cover data generally does not necessitate transformation; and on occasions, PERMANOVA results can vary depending whether transformation is applied or not. Did authors explored this issue?


Figures and Tables
Table 4B, D- Nice graph, but not particularly intuitive to read.

---

## Round 0.2 · accepted · Accept

Thank you for the thorough and thoughtful response to referee comments, especially those regarding your statistical design and rationale for the choices you made in each case. I agree with your responses and feel that you have addressed the comments of the referees sufficiently in the revised manuscript that I can now move this forward into production. Congratulations, and thank you for selecting PeerJ as the outlet for your work.